# Cancer-Associated Fibroblast Mediated Inhibition of CD8+ Cytotoxic T Cell Accumulation in Tumours: Mechanisms and Therapeutic Opportunities

**DOI:** 10.3390/cancers12092687

**Published:** 2020-09-21

**Authors:** Patrick Freeman, Ainhoa Mielgo

**Affiliations:** Department of Molecular and Clinical Cancer Medicine, First floor Sherrington Building, University of Liverpool, Liverpool L69 3GE, UK; hlpfreem@liverpool.ac.uk

**Keywords:** CAF, cytotoxic T cell, T cell recruitment, T cell infiltration, T cell function

## Abstract

**Simple Summary:**

The ability of the immune system to kill tumour cells is a natural and extremely effective defence mechanism for fighting cancer. Cytotoxic-T-cells are a critical component of our immune system which function is to eliminate cancer cells. In some cancers, especially those with a rich tumour stroma, these cytotoxic-T-cells are unable to reach and kill the tumour cells. Cancer-associated fibroblasts are the most abundant cells in the tumour stroma and play a key role of the recruitment, infiltration and function of cytotoxic T-cells in the tumour, via several molecular mechanisms which we describe in this review.

**Abstract:**

The tumour microenvironment (TME) is the complex environment in which various non-cancerous stromal cell populations co-exist, co-evolve and interact with tumour cells, having a profound impact on the progression of solid tumours. The TME is comprised of various extracellular matrix (ECM) proteins in addition to a variety of immune and stromal cells. These include tumour-associated macrophages, regulatory T cells (Tregs), myeloid-derived suppressor cells, as well as endothelial cells, pericytes and cancer-associated fibroblasts (CAFs). CAFs are the most abundant stromal cell population in many tumours and support cancer progression, metastasis and resistance to therapies through bidirectional signalling with both tumour cells and other cells within the TME. More recently, CAFs have been shown to also affect the anti-tumour immune response through direct and indirect interactions with immune cells. In this review, we specifically focus on the interactions between CAFs and cytotoxic CD8+ T cells, and on how these interactions affect T cell recruitment, infiltration and function in the tumour. We additionally provide insight into the therapeutic implications of targeting these interactions, particularly in the context of cancer immunotherapy.

## 1. Introduction

Our understanding of the so-called tumour microenvironment (TME) has seen significant advancement through a large number of studies conducted over the last decade [1,2,3,4,5,6,7]. The TME describes the entirety of the components within the tumour mass, such as infiltrating immune cells and non-malignant stromal cells, in addition to the malignant cells themselves.

Cancer-associated fibroblasts (CAFs) are the dominant stromal cell population in many solid tumours [8,9,10] and form the focus of the present review. CAFs exhibit several functions in cancer: promoting malignant cell growth through bidirectional signalling with both tumour cells and other cells within the TME [11,12]; facilitating the process of metastasis through synthesis and remodelling of extracellular matrix (ECM) components and secretion of angiogenic factors such as vascular endothelial growth factor (VEGF) [11,13]; sustaining cancer cell bioenergetics through release of CAF-derived metabolites [14,15]; contributing to tumour chemoresistance [3,16]; and promoting evasion of immune surveillance [17,18].

However, despite the variety of CAF-mediated tumour-promoting functions, CAF ablation strategies have largely been deemed as deleterious. This was exemplified by the landmark studies of Özdemir et al. [19] and Rhim et al. [20], in which genetic depletion of CAFs in murine models of pancreatic ductal adenocarcinoma (PDAC) resulted in more aggressive tumours and worse survival outcomes. These findings are in contrast to other studies which have shown a beneficial effect of the genetic ablation of fibroblast activation protein (FAP+) CAFs on survival outcomes in preclinical models of PDAC and Lewis lung carcinoma [21,22]. It should be noted, however, that other studies have reported severe systemic toxicities such as cachexia and reduced erythropoiesis in transgenic models of PDAC and transplantable models of colorectal carcinoma upon genetic ablation of the FAP+ CAF population [23]. Therefore, the translational relevance of large-scale targeting of the CAF population remains an area of close contention.

What has been made clear through these studies is that there is significant phenotypic heterogeneity within the CAF population with certain subtypes acting to restrain, rather than promote tumour progression [24]. There is now a large body of work seeking to delineate the roles of these different CAF subtypes within the TME but this is not the topic of the present review. Identification of unfavourable CAF subpopulations will be paramount when it comes to the design of therapeutics that selectively target these tumour supporting CAFs.

However, whilst our understanding of CAF heterogeneity has improved over the last half decade, CAF subtyping is still very much in its infancy and identifying unique CAF markers of CAF subtypes has proven to be a significant challenge in the field of cancer biology. At this early stage, it is currently unknown whether this subtyping of CAF populations will confer therapeutic benefit. We believe therefore, that it is important to also further characterise the functional interactions of CAFs within the TME, alongside approaches to delineate CAF heterogeneity, for the rational design of therapeutic strategies that target the pro-tumourigenic functions of CAFs.

Studies conducted in recent years have developed a greater appreciation for the role of the immune system in regulating/deregulating cancer progression. This has also led to a surge of interest in the field of immuno-oncology, with immunotherapeutic strategies such as adoptive transfer of chimeric antigen receptor (CAR) T cells and immune checkpoint inhibitors (ICIs) being the subject of a large number of clinical trials over the last decade. Critically, in the context of solid tumours, the efficacy of these immunotherapeutic strategies is dependent on a functional cytotoxic CD8+ T cell response within the TME [25,26]. Given that CAFs are one of the principle components of the TME in many malignancies and with the current efforts being made to improve cancer immunotherapy, the present review will focus specifically on how CAFs affect cytotoxic CD8+ T cell recruitment, infiltration and function within the tumour. Additionally, we shall explore the implications of targeting CAF-T cell interactions for cancer immunotherapy, seeking to restore CD8+ T cell cytotoxic activity and promote an effective anti-tumour immune response.

## 2. CAFs and T cells

The cancer immunity cycle proposed by Chen and Mellman describes a series of stepwise events implicit to an effective anti-tumour T cell response [27]. Critically, the efficacy of cancer immunotherapy is reliant on all stages of this cycle being intact. CAFs are able to inhibit the function of CD8+ T cells, reducing their tumour killing capacity, through interference with several stages of the cancer immunity cycle. In particular, CAFs have been shown to abrogate CD8+ T cell function through: deregulation of T cell trafficking/recruitment to the tumour; reducing T cell infiltration into the TME; and suppression of the cytotoxic function of T cells which are present within the tumour milieu (Figure 1, Table 1). The present review will provide an updated summary of the literature concerning CAF-mediated deregulation of CD8+ T cell activity through these three main mechanisms.

Given the fact that CD8+ T cells directly mediate tumour cell killing and their importance in the field of immuno-oncology, the literature detailed herein is specifically focused on the effects of CAFs on cytotoxic CD8+ T cells. Nonetheless, it should be mentioned that the pleiotropic functions of CAFs extend also to regulating the activity of other T lymphocytes such as regulatory T cells (Tregs) and CD4+ T helper cells and other immune cell populations present in the tumour microenvironment [28].

## 3. CAFs Mediated Inhibition of CD8+ T Cell Trafficking/Recruitment

The ability of the immune system to mount a successful response relies, in part, on the recruitment of various immune cells to the site of inflammation. This recruitment is dependent on a highly sophisticated system involving chemokines and their corresponding chemokine receptors, enabling a highly specific and localised immune response. Accordingly, upon their activation within the TME, CAFs can modulate the recruitment of peripheral CD8+ T cells towards the tumour through secretion of numerous cytokines and chemokines. For instance, activated CAFs are able to promote the trafficking of CD8+ T cells away from the juxta-tumoral compartment and towards the pan-stromal compartment of PDAC tumours in a chemokine-dependent manner [29]. The juxta-tumoral compartment describes the area immediately adjacent to the tumour cells themselves, whereas the panstromal compartment represents areas non-adjacent to the tumour. The resulting low juxta-tumoural CD8+ T cell infiltrate is an adverse indicator of prognosis in the context of human PDAC. Mechanistically, CAF-secretion of C-X-C motif chemokine 12 (CXCL12) guides peripheral CD8+ T cell migration towards activated CAFs located in the stromal regions surrounding the tumour. This leads to the sequestration of CD8+ T cells in the pan-stromal compartment and reduced migration into tumour islets. Notably, flow cytometric analysis revealed a greater expression of the CXCR4 (the receptor for CXCL12) on circulatory CD8+ T cells from PDAC patients compared to healthy donors [29]. Corroborating this observation, preclinical studies in an autochthonous murine model of PDAC revealed that both pharmacological inhibition of CXCR4 and genetic ablation of CXCL12 producing CAFs led to a rapid accumulation of CD8+ T cells within the tumour and reduced tumour growth [30]. A recent study investigating the heterogeneity of stromal cell populations in triple-negative breast cancer (TNBC) patients and utilising single cell RNA-sequencing, revealed the presence of an inflammatory CAF (iCAF) subpopulation [31]. Of note, these iCAFs displayed a strong upregulation of the CXCL12-CXCR4 chemoattractant pathway genes, and the authors demonstrated a strong association of CD8+ T cell dysfunction and exclusion with iCAF presence [31].

## 4. CAF-Mediated Regulation of CD8+ T cell Infiltration in Tumours

A key hallmark of activated CAFs is their ability to aberrantly deposit extracellular matrix (ECM) proteins such as collagen, fibronectin, hyaluronan, in addition to matrix degradation enzymes [32]. This deregulation of the ECM has long been known to facilitate excessive tumour growth, invasion and metastasis [33,34]. More recently, CAF-mediated deregulation of the ECM has been implicated in the modulation of immune cells, and in particular, CD8+ T cells, within the TME. The ability of CD8+ T cells to recognise tumour cells and exert a cytotoxic effect requires the physical interaction between the CD8+ T cells and the tumour cells. Hence, appropriate localisation and migration of CD8+ T cells towards the tumour cells within the TME is an essential prerequisite for effective T-cell mediated killing of tumour cells. Various studies have shown an implicit role for CAFs in modulating T cell infiltration into, and within the TME. Using a live cell imaging approach in viable slices of human lung tumours, a study revealed that CAFs are able to reduce the infiltration of CD8+ T cells into tumour islets, through biophysical deregulation of ECM components [35]. CD8+ T cells were found to preferentially accumulate within the stromal regions of these tumours, which are characterised by a far “looser” network of fibronectin and collagen fibers, as opposed to accumulating within tumour islets, which are encapsulated in dense networks of collagen and fibronectin parallel fibres [35]. Matrix loosening through collagenase treatment led to an increased infiltration of CD8+ T cells into tumour islets, with a twofold increase in the number of T cells in contact with tumour cells. Furthermore, this same group went on to corroborate these findings in the context of ovarian cancer, utilising the same live cell imaging approach, with CD8+ T cells accumulating within the loose-collagen regions [36]. Similarly, analysis of human PDAC tissues has revealed that high tumour activation of focal adhesion kinase (FAK) correlates with a greater deposition of CAF-derived collagen 1, which culminates in a reduced number of tumour infiltrating CD8+ T cells and predicts a poor clinical outcome [37]. Importantly, both genetic and pharmacological inhibition of FAK1 in transgenic murine models of PDAC resulted in decreased levels of fibrosis and increased T cell infiltration. This was exemplified by a reduction in collagen deposition and numbers of both total and proliferating FAP+ and α-smooth muscle action (αSMA+) CAFs, in addition to a near 5-fold increase in CD8+ T cell infiltration into PDAC tissue [37]. In corroboration with these findings, other studies have shown that CD8+ T cells preferentially accumulate in areas of low collagen density [38]. 3D-invasion assays revealed that high-density collagen matrices significantly reduce the migration distance and number of invading CD8+ T cells in in vitro models of PDAC, impeding T cell-tumour cell contact [38]. Similarly, other studies have shown that T cells cultured in 3D high-density collagen matrices display a reduced cytotoxic potential in patient derived models of melanoma [39]. This collagen-density dependent reduction in cytotoxic activity also coincided with a reduced secretion of interferon gamma (IFNγ). This study additionally utilised whole-transcriptome RNA-seq analysis to reveal that T cells cultured in high-density collagen matrices downregulate markers of cytotoxic activity whilst upregulating markers indicative of a Treg phenotype [39]. Overall, these data suggest that CAF reorganisation of the ECM architecture can act as a biophysical barrier to T cell accumulation within the tumour parenchyma, reducing the T-cell contact-dependent killing of tumour cells in different solid malignancies.

Conversely, several lines of evidence from a series of independent studies may call into question the importance of CAF-mediated ECM deregulation as a means of reducing T cell infiltration. Utilising novel computational imaging techniques, one study showed that while spatial distribution of CD8+ T cells in close proximity with tumour cells correlated with an increased overall patient survival in pancreatic cancer, the density of collagen 1 deposition and number of αSMA+ CAFs do not correlate with a reduced CD8+ T cell infiltrate [40]. In contrast to previous reports, this study showed that tumours with a greater paucity of CD8+ T cells actually exhibited diminished collagen 1 deposition in their surrounding milieu [40]. This suggests that collagen deposition and ECM remodelling may favour, rather than hinder, the process of T cell infiltration within the TME. This notion is supported by another study analysing the immune cell infiltrate of human PDAC specimens. Immunohistochemistry analysis revealed a significantly higher infiltration of CD8+ T cells in patients with a high stromal density compared to patients with a loose or moderate stromal density [41]. In agreement with these findings, a histological analysis of human colorectal carcinomas also showed that increased deposition of collagen and a greater desmoplastic reaction correlated with increased numbers of tumour infiltrating CD8+ T cells [42].

Therefore, at least in the context of PDAC and colorectal carcinoma, it still remains unclear whether CAF-mediated ECM deposition impedes or favours infiltration of CD8+ T cells into the tumour. CAFs have also been shown to negatively regulate the infiltration of CD8+ T cells into solid tumours through regulation of their secretome. Increased secretion of CAF-derived TGF-β has been shown to correlate with a reduced accumulation of CD8+ T cells within both metastatic urothelial cancer and metastatic colorectal carcinoma and a reduced response to immune checkpoint inhibitors [43,44]. In both settings, inhibition of CAF-derived TGF-β resulted in increased T cell accumulation within the tumour parenchyma and restored the efficacy of checkpoint inhibition in preclinical models, reducing metastatic burden [43,44]. Currently, however, the precise mechanisms through which TGF-β modulates the infiltration of CD8+ T cells to these tumours remain elusive.

It is important to also note the role of tumour hypoxia as another factor responsible for the reduced infiltration of CD8+ T cells in many solid tumours. CAF deposition of ECM contributes to the generation of a hypoxic TME, and in turn, the CAFs upregulate the expression of angiogenic factors such as VEGF [45,46]. Endothelial cells downregulate the expression of various cell adhesion molecules such as ICAM-1/2 and VCAM-1 in response to VEGF, a process known as “endothelial cell anergy” [47]. A reduction in cell adhesion molecules inhibits the extravasation of circulatory CD8+ T cells through the tumour vasculature and into the tumour itself, and studies have shown that a targeted reduction of hypoxia can restore the infiltration of T cells in preclinical models of prostate cancer [48]. The pleiotropic effects of hypoxia on the TME and the implications for cancer development and progression have been recently reviewed [49,50,51].

## 5. CAF-Mediated Suppression of T cell Cytotoxic Function

### 5.1. Direct Upregulation of Immune Checkpoint Molecule Expression

The immune system is well-orchestrated and in a healthy organism exists in a state of homeostasis. The immune system can be rapidly mobilised to act against invading pathogens or neoplastic cells whilst simultaneously, self-tolerance is maintained by negative regulation of cell activation—preventing the indiscriminate killing of host cells, as seen in autoimmune disorders [52]. Immune checkpoint molecules are a group of co-inhibitory and co-stimulatory molecules which serve to maintain this critical state of homeostasis. In the context of many cancers, chronically activated tumour reactive T cells are often dysfunctional or “exhausted” due to the activation of inhibitory immune checkpoint receptors, displaying a loss of effector function and proliferative potential [53,54,55,56,57]. CAFs can directly induce this exhausted phenotype in T cells through upregulation of a host of immune checkpoint ligands on their cell surface. For instance, fibroblasts derived from melanoma patient biopsies displayed an upregulation of both programmed death ligands 1 and 2 (PD-L1 and PD-L2), which bind to the PD-1 receptor, directly abrogating CD8^+^ T cell function [58]. More recently, both PD-L1 and PD-L2 have been shown to be upregulated in CAFs isolated from pancreatic cancer patients undergoing surgical resection [59]. This same study also demonstrates a role for pancreatic CAFs in promoting the expression of co-inhibitory immune checkpoint receptors in proliferating T cells, such as PD-1; T-cell immunoglobulin and mucin-domain containing-3 (TIM-3); lymphocyte-activation gene- 3 and cytotoxic T-lymphocyte-associated antigen-4 (CTLA-4), contributing to T cell dysfunction [59]. However, the underlying mechanism(s) through which CAFs promote the upregulation of inhibitory immune checkpoints on CD8+ T cells is incompletely understood. Albeit the authors show that no difference in immune checkpoint expression exists between CD8+ T cells grown in direct co-culture with CAFs, and those separated by transwell cultures [59]. This suggests that the upregulation of inhibitory immune checkpoints is mediated through CAF secretion of soluble factors. Importantly, high expression of PD-1 on tumour-infiltrated lymphocytes is also widely known as an adverse indicator of prognosis in PDAC [60]. Interestingly, recent studies have shown that CAFs isolated from murine lung adenocarcinomas and melanoma tumours can directly induce apoptosis in tumour-specific CD8^+^ T cells through simultaneous upregulation of PD-L2 and Fas ligand, leading to increased tumour survival [61]. Strikingly, this study also shows that certain CAFs may also be able to participate in antigen presentation, leading to direct killing of tumour-reactive CD8+ T cells in an antigen-specific manner through engagement of PD-L2 and Fas ligand [61].

### 5.2. Indirect Upregulation of Immune Checkpoint Molecule Expression

In addition to upregulation of immune checkpoint molecules on their own cell surface, CAFs can also act indirectly through paracrine signalling interactions to upregulate the expression of immune checkpoint molecules on both tumour cells and other cells of the TME. For instance, CAF-derived CXCL5 was shown to upregulate the expression of PD-L1 in murine melanoma and colorectal carcinoma cell lines upon activation the CXCR4 in a PI3K-AKT-dependent manner [62]. Similarly, patient-derived hepatocellular carcinoma CAFs were shown to upregulate the expression of PD-L1 on the surface of neutrophils, another prominent component of the TME, through secretion of IL-6 and activation of a STAT3-dependent signalling pathway [63]. In addition, these CAF-primed neutrophils were then able to suppress T cell proliferation and production of IFNγ in a PD-L1-dependent manner [63].While less is known about CAF induction of immune checkpoint expression on other cells of the TME, it is important to note that CAFs mediated deregulation of CD8+ T cell function is not limited to direct interactions between these two cell types.

### 5.3. T Cell Suppression through CAF Metabolic Reprogramming

The role of CAF metabolic reprogramming in promoting tumour progression and contributing to therapeutic resistance has now been well established. For instance, paracrine transfer of CAF metabolites such as deoxycytidine to tumour cells contributes to gemcitabine resistance in the context of PDAC [64]. A more comprehensive overview of CAF metabolic reprogramming in cancer can be found in several recent review articles [65,66]. Accumulating evidence has now revealed that CAF metabolic reprogramming serves as another mechanism for the suppression of cytotoxic T cell function within the TME. For example, tumour-educated mesenchymal stromal cells (MSCs) isolated from human cervical cancer (CeCa) patients were shown to suppress the proliferation, activation, and effector functions of CD8+ T cells through production of purinergic nucleosides [67]. Bone-marrow-derived MSCs are one of the progenitor populations which give rise to CAFs [68,69], of which there are many functionally and phenotypically diverse cellular sources [70]. Mechanistically, CeCa-derived MSCs were shown to upregulate the expression of the ectonucleotidases CD39 and CD73, which catalyse the generation of the immunosuppressive nucleoside adenosine through hydrolysis of ATP, ADP, and AMP [67]. Adenosine then exerts its inhibitory effect on cytotoxic T cell activity through binding to the A2A receptor [67]. The role of adenosine in regulating tumour immunity and a summary of recent clinical strategies to therapeutically target this pathway has been recently reviewed [71].

Another example of T cell suppression caused by CAF metabolic reprogramming has recently been observed using CAFs derived from prostate cancer patients [72]. This study identifies glycolytic CAFs as a major source of extracellular lactate, at least in the context of prostate cancer, with the immunosuppressive effects of lactic acidosis on cytotoxic T cell activity within the TME being well documented [73,74,75,76]. In addition, it has been shown that CAF expression of the enzyme arginase II is able to inhibit the infiltration and proliferation of T cells in the context of pancreatic cancer, acting as an indicator of poor prognosis [77]. Sufficient levels of extracellular arginine are crucial for T cell activation and proliferation [78]. Therefore, an increased expression of CAF arginase II leads to a reduced concentration of arginine within the TME, due to an increased hydrolysis of arginine to ornithine and consequently, a decrease in T cell proliferation and cytotoxic function.

### 5.4. Interference with Antigen Presentation and T Cell Activation

Cross-presentation of tumour-specific antigens by antigen-presenting cells (APCs) such as dendritic cells (DCs) is critical for the generation of antigen-specific T cells and a sustained anti-tumour immune response. The release and cross-presentation of tumour antigens to prime CD8+ T cells is the integral step in the initiation of the “Cancer Immunity cycle” described by Chen and Mellman [27]. Therefore, interference with antigen presentation and/or suppression of professional APCs can indirectly suppress the cytotoxic function of CD8+ T cells. In this regard, recent studies have shown that CAFs are able to interfere with various aspects of DC biology, including maturation and trafficking through the secretion of several factors. For instance, through secretion of IL-6, CAFs derived from human hepatocellular carcinoma are able to upregulate the activation of the STAT3 signalling pathway in DCs, which in turn, induces a regulatory DC phenotype which is unable to prime and activate T cells [79]. In comparison to normal mature DCs, the CAF-educated regulatory DCs were shown to have a reduced expression of antigen-presenting and co-stimulatory molecules (CD1a, HLA-DR, CD80 and CD86). As a result, CD8+ T cells were functionally impaired, displaying both a lower expression of IFNγ and a lower proliferative potential when grown in co-culture with regulatory DCs [79]. Of note, other studies have shown that T-cell-conditioned media are able to upregulate IL-6 production in CAFs isolated from lung tumour biopsies [80]. While these observations were made in separate cancer types, this highlights an example of reciprocal signalling between these two cell types and further studies should ascertain whether a negative feedback mechanism is in place, involving T cell secretory factors, subsequent release of IL-6 by CAFs and finally, induction of the regulatory DC phenotype.

There is also evidence that CAFs may act directly on CD8+ T cells to interfere with the process of antigen recognition and activation, through inhibition of the T cell receptor (TCR). Utilising both in vitro and syngeneic orthotopic murine models of PDAC, a study showed a suppressive effect of CAF-derived transforming growth factor beta-induced protein (TGF-βi) on the antigen-specific activation of ovalbumin specific CD8+ T cells [81]. Mechanistically, TGF-βi secreted by CAFs binds to CD61 on the surface of CD8+ T cells, driving binding of HIC-5 protein to Y505 phosphorylated Lck and blunting TCR signal transduction [81,82]. This leads to reduced proliferation and activation of antigen specific CD8+ T cells, whilst TGF-βi blocking antibody restored CD8+ T cell proliferation and reduced T cell exhaustion, characterised by reduced expression of PD-1 and TIM-3 in vitro [81]. Furthermore, antibody-mediated depletion of TGF-βi led to reduced tumour burden in spontaneously developing murine models of PDAC and critically, these tumours were more highly infiltrated with CD8+ T cells that expressed greater levels of granzyme B, IFNγ and TNFα [81].

Finally, while the topic of CAF heterogeneity will be covered in greater detail in another review in this journal issue, recent evidence has revealed a novel subpopulation of CAFs with antigen-presenting capabilities. These so-called “apCAFs” were found to express MHC class II at their cell surface, whilst also possessing the capacity to present the ovalbumin peptide to CD4+ T cells ex vivo [83]. Co-stimulatory molecules such as CD80, CD86 and CD40 are utilised by professional APCs to elicit the vital secondary signal necessary for T cell clonal expansion upon TCR ligation [84]. The absence of these co-stimulatory markers within the “apCAF” population is particularly noteworthy, with the authors postulating that the MHCII present on these cells acts as a decoy receptor, deactivating CD4+ T cells either through induction of anergy or differentiation into immunosuppressive Tregs. Future investigations into “apCAFs” are required to provide greater mechanistic insight into these interactions; however, the implications are great. Increased Treg differentiation would unfavourably offset the CD8+ to Treg ratio, which predicts a poor clinical outcome and poor response to therapeutics in multiple types of cancer [85,86,87].

### 5.5. Therapeutic Strategies to Overcome CAF-Mediated Inhibition of CD8+ T Cells

Since the breakthrough approval of the immune checkpoint CTLA-4 inhibitor ipilimumab for the treatment of metastatic melanoma in 2011, the field of clinical immuno-oncology has seen rapid development and a surge of interest. The success of ipilimumab represents an important proof of concept for therapeutic modulation of the immune system, with CTLA-4 being the first target to bear translational relevance. This has led to the emergence of new and improved checkpoint inhibitors over the last decade, the use of which has been extended to an increasing number of indications for widespread clinical benefit [88]. Immune checkpoint blockade represents a paradigm shift in being able to harness the immune system to selectively target tumour cells which has also led to a renewed interest into designing anti-cancer vaccines based on mutation-associated neoepitopes [89]. In addition to this, recent advancements into CRISPR-Cas9 technology have revolutionised the generation of highly efficacious CAR-T cell-based therapies [90]. However, the success of all three of these immunotherapeutic strategies: immune checkpoint inhibitors, cancer vaccines and adoptive transfer of CAR-T cells, is reliant on a functionally active and tumour infiltrated CD8+ T cell population.

Currently, the use of these immunotherapeutic strategies has seen the most widespread clinical benefit in the context of non-solid cancers and cancers with a non-prominent tumour stroma [91,92]. In solid tumours, which are characterised by a significant stromal reaction and abundance of CAFs, the capacity of T-cells to kill cancer cells requires successful T cells recruitment, infiltration, and cytotoxic function within the tumour. The purpose of this review is to give an overview of the literature concerning different mechanisms by which CAFs can interfere with these three processes and ultimately, prevent T-cell mediated tumour killing. As previously mentioned, large-scale depletion of the CAF population has largely been shown to be deleterious to patient outcomes, exemplified by the failure of clinical trials utilising this approach (NCT01130142). There is, therefore, an urgent clinical need to develop novel therapeutics strategies that take a more nuanced approach when it comes to CAF inhibition. Selective targeting of CAF functions that interfere with T cell recruitment, infiltration and/or function in tumours is likely a more sensible alternative to large-scale CAF inhibition, and represents an underappreciated clinical niche. Using a combinatory approach, inhibition of CAF functions that interfere with T cells recruitment and function in tumours could unleash the full potential of immunotherapies such as ICIs and CAR-T cells extending their use into the treatment of solid tumours.

Several strategies to overcome the inhibitory effects of CAFs on T-cells are beginning to emerge; however, the field is still very much in its infancy. As previously discussed, IL-6 and CXCL12 are two CAF-derived factors that impair T-cell function and recruitment to the TME [29,79]. Accordingly, the clinical translatability of specific IL-6 and CXCL12 inhibitors has been evaluated in preclinical models of melanoma and colorectal carcinoma [93,94]. In both studies, blockade of these CAF-derived factors showed synergistic effects with anti-PD-1/PD-L1 therapy and increased the infiltration of CD8+ T cells within the TME. Interestingly, transcriptional profiling of these murine melanoma tumours revealed an elevation of both cxcl9 and cxcl10 gene signatures upon combinatory blockade of IL-6 and PD-L1 [93]. These chemokines are known to orchestrate CD8+ T cell recruitment, and their lack of expression is associated with limited infiltration of antigen specific T cells in various cancers [95] including melanoma [96,97], dictating efficacy of adoptive T cell transfer in the latter [98]. Therefore, selective blocking of CAF derived IL-6 in particular may represent a clinically relevant therapeutic approach, with the potential to ameliorate both immune checkpoint inhibition and adoptive T cell therapy through enhanced CD8+ T cell recruitment. Similarly, a recent study revealed that the pharmacological blockade of NOX4, a ROS-producing enzyme with established roles in CAF activation [99], sensitised murine models of lung, breast and colorectal cancer to both anticancer vaccination and anti-PD-1 checkpoint inhibition through enhanced CD8+ T cell infiltration [100]. Importantly, Setanaxib, the selective small molecule inhibitor of NOX-4 has been subject to clinical evaluation through phase II in the context of liver fibrosis (NCT03226067), displaying a favourable safety profile; this compound could therefore be feasibly repurposed for treatment of CAF-rich, immune-excluded tumours.

Overall, however, there is currently a rather limited body of work focused on therapeutically exploiting CAF-mediated deregulation of CD8+ T cell function with this area of research needing further attention in future studies.

## 6. Conclusions and Future Perspectives

There is now a growing body of evidence detailing the mechanisms through which CAFs can negatively regulate the activity of cytotoxic CD8+ T cells within the TME of solid malignancies. As presented herein, this CAF mediated regulation of T cell function is mediated through interference with T cell-recruitment, infiltration, and cytotoxic activity. Future studies must seek to further characterise these molecular interactions between CAFs and CD8+ T cells. In particular, greater emphasis needs to be made into delineating the precise role of CAF production of ECM components in regulating T cell infiltration in the TME. As described, the jury is still out on whether deposition of ECM serves to facilitate T cell infiltration or whether it forms a barrier to biophysically impede the movement of T cells, as canonically believed to be the case.

Furthermore, future efforts must be made to build upon the basic science of CAF-T cell biology presented within this review to facilitate the rational development of therapeutic strategies which seek to target these interactions. Additionally, previous studies already showed that large-scale genetic ablation of the CAF population as a whole is deleterious and translates to increased tumour aggressiveness and reduced survival in preclinical studies [19,20], owing to the significant complexity of CAF heterogeneity which is still far from being fully comprehended. While the search to better understand CAF phenotypic heterogeneity is underway, we should simultaneously focus our efforts into improving the clinical translatability of targeting CAF-T cell interactions. In theory, this will also serve to ameliorate current immunotherapeutic treatment strategies such as adoptive T cell transfer, immune checkpoint inhibitors and anticancer vaccination, extending their use in the treatment of a wider range of solid malignancies.

## Figures and Tables

**Figure 1 cancers-12-02687-f001:**
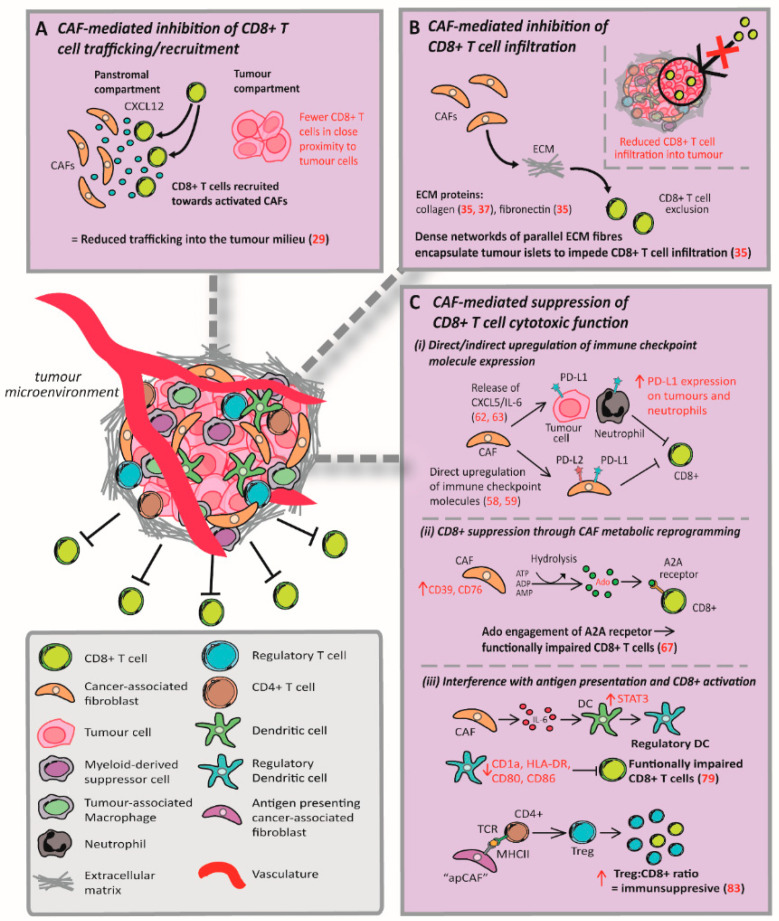
Mechanisms of cancer-associated fibroblast inhibition of CD8+ cytotoxic T cells. CAFs mediate inhibition of CD8+ T cells through interference with (**A**) T cell-trafficking/recruitment, (**B**) infiltration, and (**C**) cytotoxic function. Figure compiled from the following sources: [29,35,37,58,59,62,63,67,79,83].

**Table 1 cancers-12-02687-t001:** Summary of CAF-derived molecules/factors responsible for the inhibition of CD8+ T cells through interference with T cell-trafficking/recruitment, infiltration and cytotoxic function.

CAF Derived Molecule	Effect on CD8+ T Cells	Tumour Type/Model	Type of Analysis Performed	Result of Study	Reference
CXCL12	Decreased recruitment. CD8+ T cells sequestered in panstromal compartment	Pancreatic cancer. TMA analysis of patient tissue. Transgenic KPC murine model. In vitro migration assays utilising circulatory patient derived CD8+ T cells.	Tissue microarray analysis of patient tissues and IHC analysis of KPC tumours. Migration assays assessed differential migration of CD8+ T cells towards activated pancreatic stellate cell (aPSC) conditioned media (CM).	Lower CD8+ T cells in juxtatumoral compartment. CD8+ T cells migrate preferentially towards aPSC CM. Knockdown of CXCL12 = reduced CD8+ T cell migration	Ene-Obong et al., 2013 [29]
TGF-β	Decreased infiltration + reduced response to immune checkpoint inhibitors	mUC and breast cancer. Transcriptome RNA-seq of mUC patient tissue. Syngeneic orthotopic transplantation of EMT6 murine breast cancer cells.	RNA-sequencing of 298 tissue samples, derived from mUC patients treated with anti-PD-L1 therapy. Flow cytometric and quantitative IHC analysis of EMT6 tumours. Mice treated with single agent/combination treatment anti-PD-L1 and anti-TGF-β.	Lack of response to anti-PD-L1 associated with low CD8+ T cell infiltration and a signature of TGF-β signalling in fibroblasts. Reduced tumour burden and increased CD8+ infiltration a under combinatory anti-PD-L1 and anti-TGF-β treatment.	Mariathasan et al., 2018 [43]
Collagen and fibronectin	Decreased infiltration	Lung cancer. Live imaging of viable human lung cancer slices.	2-photon imaging and second harmonic generation (SHG) + immunostaining, to visualise collagen and fibronectin density.	Reduced infiltration of CD8+ T cells in regions of dense collagen/fibronectin.	Salmon et al., 2012 [35]
		Pancreatic cancer. Analysis of patient tissue. Pancreatic cancer	IHC analysis of PDAC patient tissue using picrosirius red to measure collagen deposition.	Activation of FAK1 = increased deposition of collagen and reduced numbers of tumour infiltrating CD8+ T cells.	Jiang et al., 2016 [37]
		3D-in vitro invasion assays	Brightfield microscopy assessed depth of T cell migration into collagen matrices of varying densities containing SDF-1 (CXCL12) as chemoattractant.	Higher density collagen matrices impeded invasion of T cells towards SDF-1.	Hartmann et al., 2014 [38]
PD-L1 and PD-L2	Suppression through direct CAF upregulation of immune checkpoint expression	Melanoma. In vitro co-cultures using patient biopsy derived CAFs	T cell function assessed by measuring IFN-γ in co-cultures with melanoma patient derived CAFs. CAF expression of PD-L1 and PD-L2 assessed by flow cytometry.	IL-1 treated melanoma CAFs led to reduced T cell production of IFN-γ in co-cultures and increased CAF expression of both PD-L1 and PD-L2.	Khalili et al., 2012. [58]
PD-L2 and FasL	Suppression of activity + direct induction apoptosis.	Lung cancer and melanoma. CAFs isolated from genetically engineered LSL mouse model of lung cancer and syngeneic orthotopic transplantation of B16.F10 melanoma cells.	Co-culture cytotoxicity assays with CAFs from lung/melanoma tumours and ovalbumin-specific OT-I T cells, and OVA-expressing tumour cells. Analysed by flow cytometry.	CAF-conditioned OT-I T cells display reduced tumour killing capacity. CAFs cause antigen-dependent decrease in OT-I T cell viability due to increased expression of PD-L2 and FasL	Lakins et al., 2018. [61]
CXCL5	Suppression through induction of PD-L1 expression in tumour cells	Melanoma and colorectal carcinoma. In vitro models of melanoma and colorectal cancer cell lines, primary CAFs from subcutaneous xenograft melanoma tumours	Co-culture assays of tumour cell lines and primary CAFs analysed by flow cytometry. qRT-PCR, IF and IB reveal CXCL5 overexpression in CAFs.	Co-culture induced overexpression of PD-L1 in tumour cells in a CXCL5 dependent manner.	Li et al., 2019 [62]
IL-6	Suppression through induction of PD-L1 expression on neutrophils.	Hepatocellular carcinoma (HCC). In vitro assays neutrophils cultured in HCC patient derived CAF conditioned media.	Neutrophils cultured in CAF-conditioned media and analysed for upregulation of PD-L1 by qRT-PCR and flow cytometry. CAF-primed neutrophils co-cultured with purified peripheral T cells.	HCC CAF-conditioned media upregulates expression of PD-L1 in neutrophils in an IL-6 dependent manner. CAF-primed neutrophils suppress T cell proliferation and IFN-γ production.	Cheng et al., 2018 [63]
	Suppression through generation of immunosuppressive regulatory DCs.	Hepatocellular carcinoma. In vitro co-culture assays utilising CAFs derived from HCC patient tumour explants and DCs differentiated from healthy CD14+ monocytes.	IL-6 neutralising Abs added to co-culture systems. Analysis by flow cytometry. CAF-educated DCs co-cultured with peripheral blood lymphocytes. Analysis by flow cytometry.	CAF-educated DCs exhibit more immunosuppressive (regulatory)phenotype. Induction of this phenotype is IL-6 dependent. CAF-educated DCs inhibit T cell proliferation and CD8+ production of IFNγ.	Cheng et al., 2016 [79]
CD39/CD73	Suppression through increased generation of immunosuppressive adenosine	Cervical cancer. In vitro assays involving patient-derived MSCs and PBMC derived CD8+ T cells.	MSC CD39/CD73 expression determined by flow cytometry and IHC. Ado containing MSC conditioned media added to CD8+ T cells, proliferation determined by colorimetric assay, IFN-γ measured by flow cytometry.	CD39/CD73 expression confirmed and were confirmed to hydrolyse ATP, ADP and AMP to form Ado. Ado containing MSC conditioned media strongly inhibited proliferation and activation of CD8+ T cells.	de Lourdes Mora-García et al., 2016 [67]
TGF-βi (βig-h3)	Suppression through blunting TCR signal transduction	Pancreatic cancer. In vitro assays using CAFs derived from transgenic murine models of PDAC. In vivo syngeneic subcutaneous transplantation of transgenic murine KC cells.	CD8+ T cells cultured in CAF-conditioned media (confirmed to contain βig-h3 by ELISA) +/- βig-h3 depleting antibody. Proliferation and expression of exhaustion markers in CD8+ T cells measured by flow cytometry. Confocal microscopy used to determine βig-h3 interaction with CD61 on CD8+ T cells. Flow cytometry used to characterise effect of βig-h3 depletion in vivo.	CAF-conditioned media (confirmed to contain βig-h3) caused reduced proliferation of CD8+ T cells and βig-h3 depletion reduced expression of exhaustion markers (PD-1, TIM-3). Confocal microscopy confirmed interaction between βig-h3 and CD61 (colocalization). βig-h3 depletion in vivo reduced tumour growth and enhanced the number of activated CD8+ T cells.	Goehrig et al., 2019 [81]

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
