# Peer review of "Cancer-Associated Fibroblast Mediated Inhibition of CD8+ Cytotoxic T Cell Accumulation in Tumours: Mechanisms and Therapeutic Opportunities"

_cancers, 2020, doi:10.3390/cancers12092687_

Round 1

Reviewer 1 Report

This review discusses the role of cancer-associated fibroblast (CAFs) in tumor immunosuppression, focusing on CD8+ T cells. In general, it is well structured and provides an interesting summary of the literature, although maybe too focused on pancreas cancer, and it would be interesting to mention what is known in other tumor types. The following points should be also addressed:

  1. The distinction between CD8+ T cell trafficking/recruitment and infiltration in tumors is not very clear, and in some cases both functions overlap. This should be clarified
  2. The majority of the abstract is a verbatim copy/paste of the beginning of the Introduction, and should be different and a bit more elaborated.
  3. The authors claim that CAF ablation strategies result in more aggressive tumors but they omit studies showing the opposite. For example, Kraman, M. et al (2010). Science330, 827–830 (2010), and Lo, A. et al. (2017). JCI Insight2:e92232
  4. The juxta-tumoral compartment and pan-stromal compartment referred to in the following sentence should be explained: “For instance, activated CAFs are able to promote the trafficking of CD8+ T cells away from the juxta-tumoral compartment and towards the pan-stromal compartment of PDAC tumours in a chemokine dependent manner.”
  5. Ref 28 and 29 are discussed in the context of trafficking/recruitment, stating that CAF secretion of TGF-β correlates with reduced accumulation of CD8+ T cells in metastatic tumors, but the two papers refer to the role of TGF-β in T cell infiltration. Also, at the end of this paragraph: “…… through deregulation of the CAF secretome”, should be: “through regulation of the CAF secretome”.
  6. Regarding CAF mediated regulation of CD8+ T cell infiltration, what about the role of the ECM deposition in CD8+T cell infiltration in solid tumors other than PDAC?
  7. The authors should comment on how hypoxia affects CD8+ cell T cell infiltration in tumors?
  8. In the section “T cell suppression through CAF metabolic reprogramming”, the following sentence is out of scope and should be removed: Another example of T cell suppression caused by CAF metabolic reprogramming has recently been observed using CAFs derived from prostate cancer patients (59). This study shows that glycolytic CAFs can produce lactate which polarises CD4+ T cells from a Th1 phenotype to a FoxP3+ 226 regulatory T cell (Treg) phenotype (59), the abundance of which in the intratumoral compartment is known to confer poor prognosis in the majority of solid tumours (60-62). The immunosuppressive effects of CAFs on CD4+ T cells are beyond the overall scope of the present review, which is focused primarily on CAF-CD8+ T cell interactions
  9. Figure 1 and Table 1 are not cited in the text.
  10. Fig1 is way too small and impossible to read, especially panel C. It should be made much bigger or the information split into more than one figure.

Author Response

  1. The distinction between CD8+ T cell trafficking/recruitment and infiltration in tumors is not very clear, and in some cases both functions overlap. This should be clarified.

Response: We have now moved the section referring to the role of CAF-derived TGF-β from section 3 to section 4, according to your later comment (comment 5). Hopefully this allows for more of a distinction between recruitment and infiltration – as we agree with the comment that these papers concerning TGF-β are more referring to its effects on CD8+ infiltration.

  1. The majority of the abstract is a verbatim copy/paste of the beginning of the Introduction, and should be different and a bit more elaborated.

Response: The abstract/introduction have been updated – removing the sentences from the introduction which overlapped with those in the abstract. The introduction now comes to the topic of CAFs far sooner and more focused. Lines: 10-39.

  1. The authors claim that CAF ablation strategies result in more aggressive tumors but they omit studies showing the opposite. For example, Kraman, M. et al (2010). Science330, 827–830 (2010), and Lo, A. et al. (2017). JCI Insight2:e92232

Response: This is a very good point, and changes have been made to accommodate evidence from both side of the debate. We have also highlighted the potential pitfalls of targeting FAP+ CAFs in particular, citing Roberts, EW. et al (2013) who give evidence for severe toxicities associated with targeting this population of cells. Lines 52-58.

  1. The juxta-tumoral compartment and pan-stromal compartment referred to in the following sentence should be explained: “For instance, activated CAFs are able to promote the trafficking of CD8+ T cells away from the juxta-tumoral compartment and towards the pan-stromal compartment of PDAC tumours in a chemokine dependent manner.”

Response: A sentence has been added to describe the meaning of these terms according to the definitions given by Ene-Obong, A. et al (2013). Lines 110-112.

  1. Ref 28 and 29 are discussed in the context of trafficking/recruitment, stating that CAF secretion of TGF-β correlates with reduced accumulation of CD8+ T cells in metastatic tumors, but the two papers refer to the role of TGF-β in T cell infiltration. Also, at the end of this paragraph: “…… through deregulation of the CAF secretome”, should be: “through regulation of the CAF secretome”.

Response: Upon re-evaluation, we agree that these references fit better in the section describing CAF mediated regulation of CD8+ T cell infiltration in tumours. Lines 195-202.

  1. Regarding CAF mediated regulation of CD8+ T cell infiltration, what about the role of the ECM deposition in CD8+T cell infiltration in solid tumors other than PDAC?

Response: References have been added to cover the role of ECM deposition/CD8+ T cell infiltration in the models of other cancers such as ovarian cancer (Bougherara, H. et al (2015), Lines 156-158) and melanoma (Kuczek, DE. et al (2019), Lines 169-175).

  1. The authors should comment on how hypoxia affects CD8+ cell T cell infiltration in tumors?

Response: A brief section has been added to describe the role of hypoxia in influencing CD8+ T cell infiltration, and we have cited several review articles to direct the reader to more comprehensive overviews on the topic of hypoxia and its effects on the TME. Lines 203-212.

  1. In the section “T cell suppression through CAF metabolic reprogramming”, the following sentence is out of scope and should be removed: Another example of T cell suppression caused by CAF metabolic reprogramming has recently been observed using CAFs derived from prostate cancer patients (59). This study shows that glycolytic CAFs can produce lactate which polarises CD4+ T cells from a Th1 phenotype to a FoxP3+ 226 regulatory T cell (Treg) phenotype (59), the abundance of which in the intratumoral compartment is known to confer poor prognosis in the majority of solid tumours (60-62). The immunosuppressive effects of CAFs on CD4+ T cells are beyond the overall scope of the present review, which is focused primarily on CAF-CD8+ T cell interactions

Response: This sentence has been removed accordingly. Lines 278-284.

  1. Figure 1 and Table 1 are not cited in the text.

Response: Figure 1 and Table 1 are now cited at line 93.

  1. Fig1 is way too small and impossible to read, especially panel C. It should be made much bigger or the information split into more than one figure.

Response: The figure has been updated and is now in portrait orientation, allowing its size to be scaled up to fit the page.

Reviewer 2 Report

In this manuscript, Freeman and Mielgo give an overview of how CAFs controls the function of CD8+ cytotoxic T cells in tumours. The Manuscript is well organised, comprehensive, but the text could be more succinct. It covers the relevant research and it is also very timely and of broad interest to the field. The authors provide new insight into the topic being reviewed, arguments are logical.

The text and language would need to be modified. For example, the font type/size is not consistent; the first half of the abstract text has a different font than the second half. For example. Also, the title is very (too) long, and spaces are lacking. There is an excessive use of acronyms, names not used more than a few times in the text can be written out instead. The section 5.5 would benefit from being more succinct. 

Similarly, the text and shapes in figure 1 is not consistent. The current version is too small, and shows too many different colours, shapes and text fonts. I suggest to change this figure so that it becomes larger and significantly more simple, containing fewer shapes and figures. For example, it is not necessary to show nine (yellow) CD8+T cells in the top left image of the tumour for the reader to understand the principle. I suggest to make the figure more simple, and only keep information and formats required for the reader to understand it. Hence, also remove the different background color of the boxes, since it is not clear what these colours signify. Also, the light grey text in the top right box is too small. I suggest to remove all info that is not absolutely required, and move the Key on the left side to a position under the figure. 

Table 1 is a nice representation of the information, but I would suggest the authors to condense the figure as much as possible (because there is a lot of white space in the table), and align all text left. Also, the current position - just before the references - and actually, after the Manuscript text is not a good position. Instead I suggest the authors place the table in the text before the section ("6.Concluding..."). The current version of the table doesn´t allow the reader to understand which experiments were done and how. Therefore, I suggest the authors move insert a column describing this to the left side of the table, followed by a column showing the model system. After (thus, on the right side) of these columns that explain the experiments, the results obtained can be shown in a column. Also, I would advice the authors to remove the Bold text in the table, since the different fonts is a bit confusing. 

I suggest the authors replace signs such as "&" with words, and carefully check for typos and that words, shapes, symbols and figures are used in a consistent manner. 

Author Response

In this manuscript, Freeman and Mielgo give an overview of how CAFs controls the function of CD8+ cytotoxic T cells in tumours. The Manuscript is well organised, comprehensive, but the text could be more succinct. It covers the relevant research and it is also very timely and of broad interest to the field. The authors provide new insight into the topic being reviewed, arguments are logical.

The text and language would need to be modified. For example, the font type/size is not consistent; the first half of the abstract text has a different font than the second half. For example. Also, the title is very (too) long, and spaces are lacking. There is an excessive use of acronyms, names not used more than a few times in the text can be written out instead. The section 5.5 would benefit from being more succinct. 

Response:

Font type/size has been standardised.

The title has been shortened.

Attempts have been made to reduce the excessive use of acronyms. Removed: TAMs (Line 16); MDSCs (Line 16); CRC (Line 191); IHC (Line 188); LAG-3 (Line 232); TILs (Line 238); FasL (Line 242); HCC (Line 252); Ado (Line 272); ARG2 (Line 288); OT-1 (Line 319); OVA (Line 331).

In order to make section 5.5 more succinct, we have removed the section on astaxanthin, given that this study was performed in immune-compromised mice, we deemed this piece of evidence weaker than other lines of evidence (Lines 398-405).

Similarly, the text and shapes in figure 1 is not consistent. The current version is too small, and shows too many different colours, shapes and text fonts. I suggest to change this figure so that it becomes larger and significantly more simple, containing fewer shapes and figures. For example, it is not necessary to show nine (yellow) CD8+T cells in the top left image of the tumour for the reader to understand the principle. I suggest to make the figure more simple, and only keep information and formats required for the reader to understand it. Hence, also remove the different background color of the boxes, since it is not clear what these colours signify. Also, the light grey text in the top right box is too small. I suggest to remove all info that is not absolutely required, and move the Key on the left side to a position under the figure. 

Response:

We have taken on board the feedback concerning the figure and implemented the changes accordingly. The figure is now in portrait orientation – allowing it to be scaled up to fit the size of the page. The font type and size has been standardised, to improve readability. We have attempted to simplify the figure – giving the boxes a uniform background colour and removing the diagram in box C which showed the reduced killing capacity of CD8+ T cells. We tried moving the key to a position underneath the rest of the figure – but felt it fit more appropriately at the bottom left – to avoid a large area of blank space, now that the size of box C has been significantly increased.

Table 1 is a nice representation of the information, but I would suggest the authors to condense the figure as much as possible (because there is a lot of white space in the table), and align all text left. Also, the current position - just before the references - and actually, after the Manuscript text is not a good position. Instead I suggest the authors place the table in the text before the section ("6.Concluding..."). The current version of the table doesn´t allow the reader to understand which experiments were done and how. Therefore, I suggest the authors move insert a column describing this to the left side of the table, followed by a column showing the model system. After (thus, on the right side) of these columns that explain the experiments, the results obtained can be shown in a column. Also, I would advice the authors to remove the Bold text in the table, since the different fonts is a bit confusing. 

Response:

These are some very good suggestions. We have attempted to condense the figure to remove as much unnecessary white space as possible. Text aligned left and bold font removed. Table has been relocated to before section 6 as suggested. Extra columns have been added to explain the experiments conducted in greater detail and state the result of these experiments/analysis.

I suggest the authors replace signs such as "&" with words, and carefully check for typos and that words, shapes, symbols and figures are used in a consistent manner. 

Response:

“&” changed to “and” (Line 412). Document has been proof read for typos.

Reviewer 3 Report

I would like to congratulate the authors for such an interesting review. It gave me lots of ideas for my own research. Therefore, I think that it will make a valuable contribution to the field. On top of that, it was very enjoyable to read. However, there are a couple of recent publications in the field that, probably due to the very recent time in which they were published, escaped the radar of the authors. I would suggest to study them and see how they could be included within the review.

  • The first is a very recent study on CAF diversity and how iCAF presence correlated with CD8 T-cell dysfunction and exclusion: EMBO J. 2020 Aug 13;e104063. doi: 10.15252/embj.2019104063
  • Similarly there is a recent study showing that Ly6C+ SMA- pCAFs inhibited CD8+ T-cell proliferation in vitro. It would be important to include it as well. Nat Cancer 1, 692–708 (2020). doi: 10.1038/s43018-020-0082-y

Author Response

I would like to congratulate the authors for such an interesting review. It gave me lots of ideas for my own research. Therefore, I think that it will make a valuable contribution to the field. On top of that, it was very enjoyable to read. However, there are a couple of recent publications in the field that, probably due to the very recent time in which they were published, escaped the radar of the authors. I would suggest to study them and see how they could be included within the review.

  • The first is a very recent study on CAF diversity and how iCAF presence correlated with CD8 T-cell dysfunction and exclusion: EMBO J. 2020 Aug 13;e104063. doi: 10.15252/embj.2019104063
  • Similarly there is a recent study showing that Ly6C+ SMA- pCAFs inhibited CD8+ T-cell proliferation in vitro. It would be important to include it as well. Nat Cancer 1, 692–708 (2020). doi: 10.1038/s43018-020-0082-y

Response: Thank you. We are glad to hear this review inspired you.

Both papers are indeed very interesting and demonstrate roles of different CAF populations in regulating the activity of CD8+ T cells and we appreciate them being brought to our attention. The first paper has been referenced accordingly – lines 122-127. However, while the implications of the second paper are extremely important, its main focus is on understanding the heterogeneity of the different CAF populations – as opposed to understanding the functional interactions between CAFs and CD8+ T cells which is the specific focus of this review. There is another review being published in this journal issue which will be covering the subject of CAF heterogeneity/subtyping in far greater detail. The observation that Ly6C+ SMA- pCAFs inhibit CD8+ T cell proliferation is certainly very interesting, but that study does not describe the mechanism through which this occurs. Thus, we believe this particular study will fit better in the review on CAFs heterogeneity.
